# Adversarial Domain Adaptation for Cell Segmentation

**Mohammad Minhazul Haq**[1]                    MOHAMMADMINHAZU.HAQ@MAVS.UTA.EDU

**Junzhou Huang**[1]                              JZHUANG@UTA.EDU

[1] *Department of Computer Science and Engineering, University of Texas at Arlington, Arlington, TX 76019, USA*

## Abstract

To successfully train a cell segmentation network in fully-supervised manner for a particular type of organ or cancer, we need the dataset with ground-truth annotations. However, high unavailability of such annotated dataset and tedious labeling process enforce us to discover a way for training with unlabeled dataset. In this paper, we propose a network named CellSegUDA for cell segmentation on the unlabeled dataset (target domain). It is achieved by applying unsupervised domain adaptation (UDA) technique with the help of another labeled dataset (source domain) that may come from other organs or sources. We validate our proposed CellSegUDA on two public cell segmentation datasets and obtain significant improvement as compared with the baseline methods. Finally, considering the scenario when we have a small number of annotations available from the target domain, we extend our work to CellSegSSDA, a semi-supervised domain adaptation (SSDA) based approach. Our SSDA model also gives excellent results which are quite close to the fully-supervised upper bound in target domain.

**Keywords:** Cell Segmentation, Unsupervised and Semi-supervised Domain Adaptation

## 1. Introduction

Convolutional Neural Network (CNN) based approaches like Fully Convolutional Network (FCN) (Long et al., 2015), U-Net (Ronneberger et al., 2015), UNet++ (Zhou et al., 2018) give very promising results in biomedical image segmentation tasks as well as in cell segmentation problems (Sirinukunwattana et al., 2016). However, to successfully train these fully-supervised methods, we need at least a few amount of annotated data i.e., images with their corresponding pixel-level ground-truth labels (Kumar et al., 2017; Zeiler and Fergus, 2014). Unfortunately, such well-annotated datasets, even if very small-sized, are highly rare in biomedical domain. Also, collecting an unannotated dataset first, and then doing the manual labeling with the help of experts is also an expensive, time-consuming and tedious process (Xu et al., 2017; Chen et al., 2019a). How if we could train a deep CNN model for cell segmentation without any further needs for the annotations? Domain Adaptation, a subclass of Transfer Learning, provides solution in such scenarios.

A multi-level adversarial network based domain adaptation approach for semantic segmentation was proposed by Tsai et al. (2018). Hoffman et al. (2017) proposed an unsupervised domain adaptation model utilizing both of pixel-level and feature-level adaptation. Isola et al. (2017) applied conditional GAN (Mirza and Osindero, 2014) for image-to-image translation problems. Chen et al. (2019b) proposed a cross-domain consistency loss based pixel-wise adversarial domain adaptation algorithm. Zhang et al. (2018) proposed

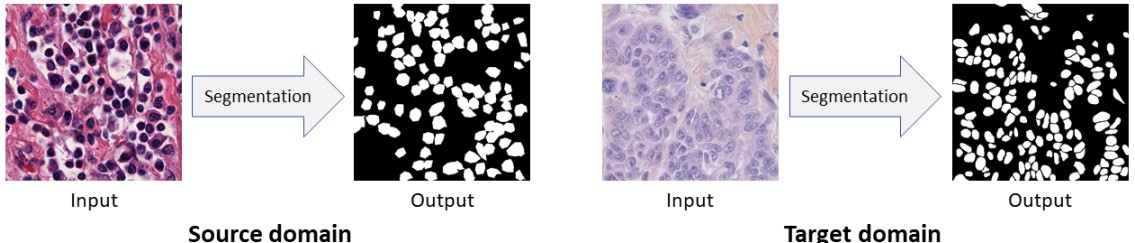

Figure 1: Images from different domains look dissimilar while their pixel-level segmentation outputs are similar. In this figure, source domain and target domain images come from Kidney Renal Clear cell carcinoma (KIRC) and Triple Negative Breast Cancer (TNBC) respectively.

a fully convolutional adaptation network for semantic segmentation. For different types of biomedical image segmentation, several adversarial network based approaches also have been proposed. A multi-connected domain discriminator based UDA model for brain lesion segmentation was proposed by Kamnitsas et al. (2017). Dong et al. (2018) introduced another UDA framework for cardiothoracic ratio estimation through chest organ segmentation. Mahmood et al. (2019) proposed a cell segmentation approach in which a large dataset is generated using synthesization. Hou et al. (2019) also synthesized annotated training data for histopathology image segmentation. Huo et al. (2018) proposed an end-to-end CycleGAN (Zhu et al., 2017) based whole abdomen MRI to CT image synthesis and CT splegonmegaly segmentation network.

In this paper, we consider the unannotated dataset, i.e. for which we want to predict the labels, as target domain. Then, with the help of another related but different annotated dataset, referred as source domain, we apply adversarial learning (Goodfellow et al., 2014) based domain adaptation technique for cell segmentation problem. Thus, our proposed framework, learns from labeled source domain and adapts to the unlabeled target domain. We very carefully observed that, images from different cell datasets, even if collected from different organs or cancer types, exhibit dissimilarity although their corresponding segmentation ground-truth labels are quite similar (see Figure 1). In summary, ground-truth labels for cell segmentation are domain-invariant.

In this work, we first propose a unsupervised domain adaptation model for cell segmentation. Because of our aforementioned observation, we apply our domain adaptation in the output space rather than in the feature space. With the help of adversarial learning, we train a robust biomedical image segmentation network to generate source-domain look-alike outputs for target images. Additionally, we use a decoder network to make target images and target predictions correlated to each other as much as possible. Finally, we extend our unsupervised domain adaptation technique to semi-supervised domain adaptation (SSDA) considering that we have some annotations available from the target domain.

Conducting extensive experiments on two cell segmentation datasets we conclude that, our proposed UDA method, CellSegUDA, outperforms both of a fully-supervised model (Ronneberger et al., 2015) trained on source domain and evaluated on target domain, and a baseline UDA model (Dong et al., 2018). Experimental result (see Section 3) also shows that,

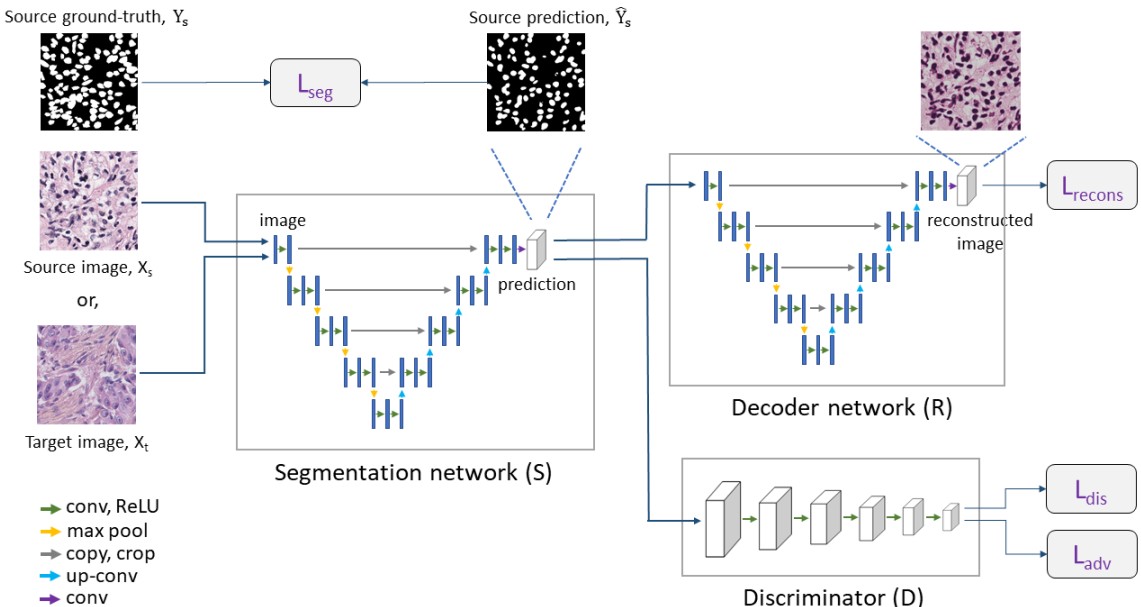

Figure 2: Complete architecture of CellSegUDA. Segmentation network generates segmentation outputs, from which decoder reconstructs input images. Discriminator distinguishes between source domain outputs and target domain outputs.

accuracy of our SSDA strategy appears very close to the upper bound of fully-supervised model trained in target domain.

Thus, the main contributions of this paper are: **1)** We propose an adversarial learning based unsupervised domain adaptation (UDA) approach to solve cell segmentation problem for unannotated datasets. **2)** Our proposed method is simple as it does not depend on any data synthesization or data augmentation. **3)** Our proposed UDA framework can be easily extended to semi-supervised domain adaptation (SSDA) in the scenario where a small portion of the target domain is labeled. **4)** Extensive and comprehensive experiments on two datasets have demonstrated the superiority of the proposed methods.

## 2. Methodology

Formally, in our cell segmentation problem, we have cell histology patches as input $X$ of size $H \times W \times 3$. Then, we want to predict the segmentation output $\hat{Y}$ of size $H \times W \times 1$. Depending on the domain, we may also have pixel-wise ground-truth label $Y$ of size $H \times W \times 1$ which is basically a binary mask.

Then, in unsupervised domain adaptation problem, we have a source domain with $N_s$ annotated images $\{(X_s, Y_s)\}$, and a target domain which has $N_t$ unannotated images $\{(X_t)\}$. In the case of semi-supervised domain adaptation problem, we assume that our target domain consists of $N_t^l$ images with annotations $\{(X_t^l, Y_t)\}$, and $N_t^u$ unannotated images $\{(X_t^u)\}$. Our ultimate goal is to learn a cell segmentation model that accurately produces the segmentation output in the target domain.

## 2.1. CellSegUDA

We refer our cell segmentation unsupervised domain adaptation (UDA) model as CellSegUDA which is shown in Figure 2. CellSegUDA consists of three modules: Segmentation network (S), Decoder (R), and Discriminator (D).

**Segmentation network (S)** Our segmentation network S takes images $X$ as input and produces the segmentation prediction $\hat{Y}$ of the same size as input, hence $\hat{Y} = S(X)$. This segmentation network can be thought as the generator module of a GAN (Goodfellow et al., 2014) framework.

We train S to generate the segmentation predictions $\hat{Y}_s$ similar to the ground-truth labels $Y_s$ in source domain. We can not compute any pixel-level loss for target predictions since ground-truth labels are not available for target images in UDA. In practice, we found dice-coefficient loss to be more effective than binary cross-entropy loss for cell segmentation tasks. Therefore, we choose dice-coefficient loss as our segmentation loss:

$$L_{seg}(X_s) = 1 - \frac{2.Y'_s.\hat{Y'}_s}{Y'_s + \hat{Y'}_s}, \tag{1}$$

where $Y'_s$ and $\hat{Y'}_s$ are flatten $Y_s$ and $\hat{Y}_s$ respectively.

Training S with only the annotated source data teaches S to make accurate predictions for source images. However, this segmentation network will generate incorrect outputs for target images as there are visual discrepancies between source images and target images. Because of our observation that cell segmentation outputs are domain-invariant, we require S to produce target domain predictions as much as close to the source domain predictions. In other words, we want to make the distribution of target predictions $\hat{Y}_t$ closer to source predictions $\hat{Y}_s$. Thus, we define adversarial loss as:

$$L_{adv}(X_t) = -\frac{1}{H' \times W'} \sum_{h',w'} \log\left(D(\hat{Y}_t)\right), \tag{2}$$

where $\hat{Y}_t = S(X_t)$, and $H'$ and $W'$ are height and width of discriminator output $D(\hat{Y}_t)$. This adverserial loss helps S to fool the discriminator so that it considers $\hat{Y}_t$ as source domain segmentation outputs.

Segmentation loss and adversarial loss altogether guides S to generate target domain predicitions $\hat{Y}_t$ which look similar to source domain ground-truths. However, it is highly probable that these target predictions are not well-correlated with corresponding target input images. The ability of reconstructing images from the predictions with similar visual appearance as input images will ensure that there is a correlation between the input image and segmentation output.

**Decoder (R)** To ensure that our target domain predictions spatially correspond to the target domain images, we use a decoder network R in CellSegUDA. In a similar way to Xia and Kulis (2017), we consider our segmentation network S as an encoder. Then, decoder R reconstructs target images from the corresponding predictions. Thus, S and R altogether works as an autoencoder.

Using our decoder network R, we first reconstruct target input images $X_t$ from $\hat{Y}_t$. Then, we calculate the reconstruction loss as:

$$L_{recons}(X_t) = \frac{1}{H \times W \times C} \sum_{h,w,c} \left(X_t - R(\hat{Y}_t)\right)^2, \tag{3}$$

where, $R(\hat{Y}_t))$ is the output of decoder for $\hat{Y}_t$, and C is the number of channels of input image X.

Thus, we minimize the following total loss while training our segmentation network:

$$L_s(X_s, X_t) = L_{seg}(X_s) + \lambda_{adv}L_{adv}(X_t) + \lambda_{recons}L_{recons}(X_t), \tag{4}$$

where, $\lambda_{adv}$ and $\lambda_{recons}$ are the weights to balance corresponding losses.

**Discriminator (D)** Since we want to generate similar predictions for both of source images and target images, we incorporate a discriminator D in CellSegUDA. This discriminator takes source domain prediction or target domain prediction as input, and then distinguishes whether the input, i.e. prediction, comes from source domain or target domain. To train D, we use following cross-entropy loss:

$$L_{dis}(\hat{Y}) = -\frac{1}{H' \times W'} \sum_{h',w'} z.\log\left(D(\hat{Y})\right) + (1 - z).\log\left(1 - D(\hat{Y})\right), \tag{5}$$

where z=0 when D takes target domain prediction as it's input, and z=1 when input comes from source domain prediction.

## 2.2. CellSegSSDA

In semi-supervised domain adaptation (SSDA) problem, we must make sure the best usages of available target domain annotations $Y_t$ while training our segmentation network S. In such scenarios, we extend our CellSegUDA framework to CellSegSSDA, a cell segmentation semi-supervised domain adaptation model.

In CellSegSSDA, for unannotated target images we do the same as CellSegUDA. However, when we encounter an annotated target data $(X_t^l, Y_t)$ while training, we additionally compute the segmentation loss $L_{seg}(X_t^l)$ in the similar manner to Equation (1). Then, while computing the total loss we incorporate $L_{seg}(X_t^l)$ so that the segmentation network learns to generate the predictions closer to target ground-truths. Therefore, Equation (4) is now modified as below:

$$L_s(X_s, X_t^l) = L_{seg}(X_s) + L_{seg}(X_t^l) + \lambda_{adv}L_{adv}(X_t^l) + \lambda_{recons}L_{recons}(X_t^l) \tag{6}$$

## 2.3. Implementations

In our work, we use U-Net (Ronneberger et al., 2015) as both of our segmentation network and decoder. We choose U-Net so that our proposed segmentation framework can be directly applied in other biomedical domains. We preferred U-Net over UNet++ (Zhou et al., 2018) because of the less number of parameters. Following DCGAN (Radford et al., 2015), we designed our discriminator consisting of five convolutional layers. To train CellSegUDA and

CellSegSSDA, we followed the training strategy from GAN (Goodfellow et al., 2014). Adam optimizer (Kingma and Ba, 2014) with learning rate 0.0001, 0.001 and 0.001 are used in segmentation network, discriminator and decoder respectively. We empirically choose 0.001 and 0.01 as $\lambda_{adv}$ and $\lambda_{recons}$ respectively. We do not use any data augmentation in our experiments.

## 3. Experiments

### 3.1. Datasets

**Dataset-1 (KIRC)** This dataset is taken from Irshad et al. (2014) in which images are extracted at 40x magnification from whole slide images (WSI) of Kidney Renal Clear cell carcinoma (KIRC). This dataset, referred as KIRC, consists of 486 H&E stained histology images of $400 \times 400$ pixel size with annotations made by expert pathologists and research fellows. In our experiments, we randomly split KIRC into 80% for training, 10% for validation and 10% for testing.
**Dataset-2 (TNBC)** Naylor et al. (2018) generated this dataset by collecting slides from Triple Negative Breast Cancer (TNBC) patients at 40x magnification. For a total of 50 H&E stained histology images of pixel size $512 \times 512$, labeling was performed by expert pathologist and research fellows. We follow the same data splitting as KIRC for this dataset which we refer as TNBC.
**Visual differences among datasets** Although both datasets consist of H&E stained histopathology images, they are collected from two different organs and different institutions. KIRC images are collected from TCGA portal (image acquiring tools are unknown to us), whereas TNBC images were acquired at Curie Institute using Philips Ultra Fast Scanner 1.6RA. Organ difference, institutional difference, and using different imaging tools and protocols cause the visual difference among the images from these two datasets. See Figure 1, where TNBC image looks dimmer than KIRC image.

### 3.2. Experimental results

**Experiment-1 (KIRC $\rightarrow$ TNBC)** In our first experiment, we choose KIRC as source domain and TNBC as target domain, denoted by KIRC $\rightarrow$ TNBC. We start with our unsupervised domain adaptation (UDA) model CellSegUDA which gives much better accuracies than a UDA baseline DA-ADV (Dong et al., 2018). We also choose a fully-supervised model U-Net (Ronneberger et al., 2015) to get an idea how it performs when directly applying transfer learning, i.e. training with only KIRC and then test it on TNBC without any modifications, which is also considered as the lower-bound of experimental performance. This poor performance of transfer learning (see the first row of Table 1) happens because of the visual domain gap between source training images and target test images, also known as domain shift problem. Figure 3(c) shows the visualization result of applying transfer learning in which many of the cells are missed out when comparing to the ground-truth. Then, training U-Net with TNBC-train and testing it on TNBC-test gives us the upper-bound (last row of Table 1). Table 1 shows that, CellSegUDA gives 6.36 higher IoU% than source-trained U-Net model. We see that, CellSegUDA also has 4.09 higher IoU% than UDA baseline DA-ADV. We check the effect of our decoder network R by training

Table 1: Segmentation results for Experiment-1 and Experiment-2. IoU denotes intersection over union. Here, unsupervised domain adaptation (UDA) baseline is denoted as DA-ADV. CellSegUDA w/o recons, CellSegUDA and CellSegSSDA refer to our proposed UDA model without reconstruction loss, proposed UDA with reconstruction loss, and proposed semi-supervised domain adaptation method respectively. CellSegSSDA(source 100% + target n%) denotes n% annotations available in TNBC-train and KIRC-train for experiment-1 and experiment-2 respectively. Results are from testing on TNBC-test and KIRC-test for experiment-1 and experiment-2 respectively.

| Method | Experiment-1 KIRC → TNBC | | Experiment-2 TNBC → KIRC | |
|---|---|---|---|---|
| | IoU% | Dice score | IoU% | Dice score |
| U-Net (source-trained) (Ronneberger et al., 2015) | 52.66 | 0.6875 | 54.82 | 0.7056 |
| DA-ADV (Dong et al., 2018) | 54.93 | 0.7079 | 55.43 | 0.7107 |
| CellSegUDA w/o recons | 56.56 | 0.72 | 56.91 | 0.7224 |
| CellSegUDA | 59.02 | 0.7394 | 57.09 | 0.7242 |
| U-Net (source 100% + target 10%) | 60.74 | 0.7534 | 56.89 | 0.7194 |
| CellSegSSDA (source 100% + target 10%) | 60.96 | 0.7557 | 58.81 | 0.7377 |
| U-Net (source 100% + target 25%) | 61.67 | 0.7607 | 59.32 | 0.7405 |
| CellSegSSDA (source 100% + target 25%) | 62.94 | 0.771 | 59.73 | 0.7443 |
| U-Net (source 100% + target 50%) | 56.73 | 0.7208 | 59.95 | 0.7464 |
| CellSegSSDA (source 100% + target 50%) | 63.59 | 0.7748 | 60.32 | 0.7494 |
| U-Net (source 100% + target 75%) | 59.06 | 0.7394 | 61.63 | 0.7592 |
| CellSegSSDA (source 100% + target 75%) | 64.96 | 0.7862 | 61.01 | 0.7541 |
| U-Net (target-trained) | 66.57 | 0.7985 | 62.04 | 0.7621 |

CellSegUDA without reconstruction loss, denoted as CellSegUDA w/o recons in Table 1. We find that, reconstruction loss really makes our segmentation network more accurate (see Figure 3(e)-(f) for visualization). Figure 3(g) also shows that we can reconstruct input images using our decoder from corresponding segmentation prediction, thus we believe that our prediction is well-correlated with its input.

Then, we assess our semi-supervised domain adaptation method CellSegSSDA for KIRC → TNBC. Source dataset, KIRC, is the same as UDA experiments. However, now we treat TNBC as partially labeled. We train CellSegSSDA considering 10%, 25%, 50% and 75% images from TNBC-train dataset has annotations available. Then, testing on TNBC-test gives us increasing IoUs and dice scores. This happens because more true positive cells can be identified and some false positive cells can be removed by CellSegSSDA as we train it with more target annotations (see Figure 3(h)-(j)). We observe that, the accuracy of CellSegSSDA approaches to the upper-bound (only lower by 1.61 IoU%) as we train with more annotations from target domain. We also compare CellSegSSDA with fully-supervised model U-Net to demonstrate the superiority of our SSDA model. This time, to train U-Net, we combine full KIRC dataset with the same 10%, 25%, 50% and 75% of TNBC-train we chose to train CellSegSSDA. As domain adaptation helps to reduce the domain shift prob-

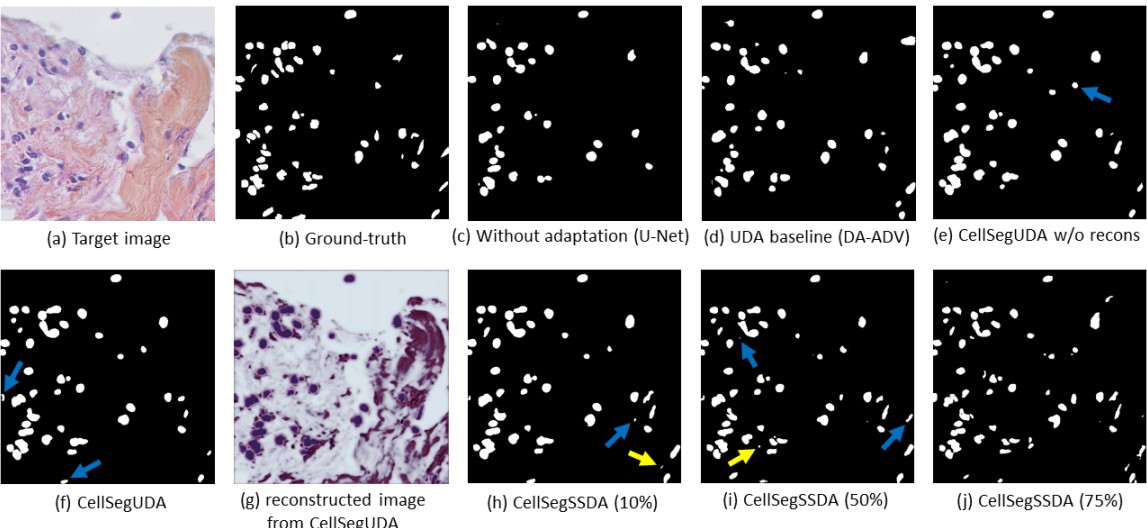

Figure 3: Visualization of segmentation for KIRC→TNBC. (g) shows that reconstructed target image (output from decoder) is quite similar to the input image which proves the efficacy of our proposed network. In (e)-(f) and (h)-(i), blue arrows indicate some missing cells of previous method. In (h) and (i), yellow arrows indicates false positives which are removed by following CellSegSSDA(50%) and CellSegSSDA(75%) respectively. Figure shows that, CellSegSSDA can identify more cells as the percentage of available annotations increases. This average-dense cell histopathology image in (a) is chosen so that the reader can easily find out the visual differences without further zooming-in.

lem, we see that CellSegSSDA outperforms fully-supervised model in all of the cases.

**Experiment-2 (TNBC → KIRC)** We conduct another experiment in the similar way to Experiment-1 by selecting TNBC as source and KIRC as target domain. This experiment also reflects the excellence of CellSegUDA and CellSegSSDA compared to other approaches in terms of segmentation accuracies (see last two columns of Table 1). Similar to experiment-1, we also see that segmentation accuracies of CellSegSSDA increase as more target images are annotated. Segmentation visualization from this experiment is shown in Figure 4. From this experiment, we once again observe that CellSegUDA performs better than CellSegUDA w/o recons which proves the validity of our decoder and the effectiveness of reconstruction loss (see reconstructed image in Figure 4(g)).

## 4. Conclusion

In this work, utilizing adversarial learning we propose a novel unsupervised domain adaptation (UDA) framework for segmenting cells in unannotated datasets. Prominent experimental results validate the effectiveness of our UDA model. Finally, assuming we have a few annotations available, we extend our work to semi-supervised domain adaptation (SSDA). To make our UDA model further accurate, we are planning to generate and utilize pseudo

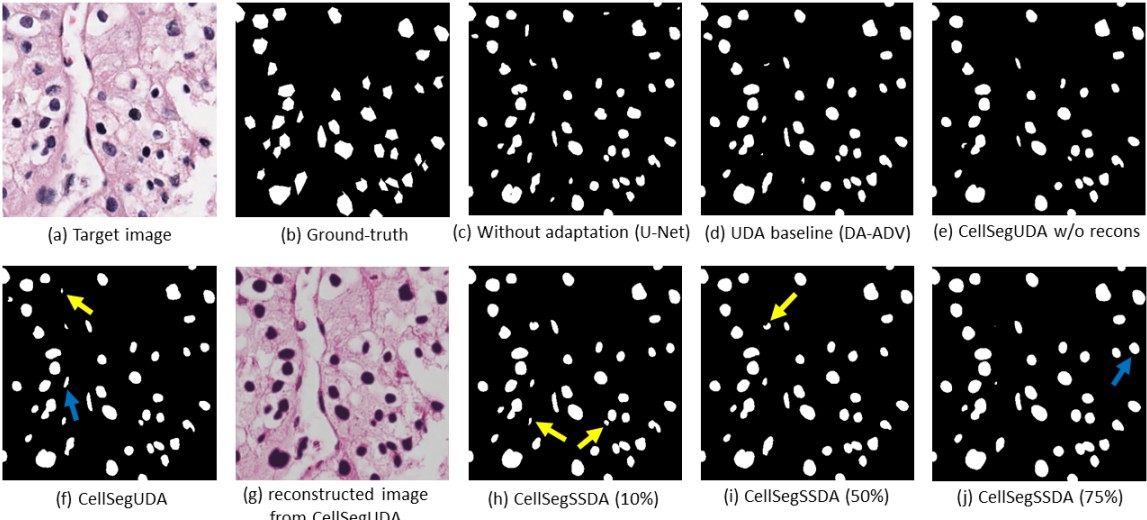

Figure 4: Visualization of segmentation for TNBC→KIRC. In (f) and (j), blue arrows indicate missing cells of previous method. In (f) and (h)-(i), yellow arrows indicate a false positive which is removed by following method. Similar to Figure 3, we chose this average-dense cell histopathology image for readability purposes.

ground-truth masks for target domain in future. We expect our proposed UDA and SSDA approach to be very useful in other biomedical image segmentation tasks.

## Acknowledgments

This work was partially supported by US National Science Foundation IIS-1718853, the CA-REER grant IIS-1553687 and Cancer Prevention and Research Institute of Texas (CPRIT) award (RP190107).

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
