# OpenReview forum: "Adversarial Domain Adaptation for Cell Segmentation"
_MIDL.io/2020/Conference — MIDL 2020_

### Official Review · AnonReviewer2 · 2020-02-25
**domain adaptation for cell segmentation**

**Rating:** 3
**Confidence:** 5
**Recommendation:** Poster

**Summary:**

CellSegUDA is proposed to perform domain adaptation for cell segmentation.
Additional data synthesization or data augmentation are not required.
The quantitative and qualitative results are promising.
The method could be extended to semi-supervised domain adaptation (SSDA).
This model can be applied to other cell modalities.


**Strengths:**

Unsupervised domain adaptation was proposed to make the algorithm salable.
The paper is easy to follow, with a nice method figure.
Both quantitative and qualitative results are provided.
Adversarial domain adaptation for cell segmentation is a good application.


**Weaknesses:**

It is not clear why the weights of lambdas are small (i.e., 0.01)
The paper listed comprehensive prior works, but actually did not compare with them except the basic U-Net and DA-ADV.
Only very similar domains are evaluated for this method.

**Justification Of Rating:**

The results are promising with a good clinical application.
The paper is well organized and wrote with easy to read figures.
The ablation tests for the % of images are comprehensive.
Ground-truth labels for cell segmentation are modeled as domain-invariant.

**Paper Type:**

methodological development

**Special Issue:**

yes

---

> ### Author Response · Authors · 2020-03-27
> **Response to Reviewer2**
>
> We would like to thank you for the positive remarks, valuable comments and feedback. Below we address the points raised.
>
> Comment 1: It is not clear why the weights of lambdas are small (i.e., 0.01)
>
> Response to Comment 1: We apologize for the confusion regarding smaller values of weights of lambdas. In our paper, the smaller values of lambda weights are chosen empirically. Following is the experimental result of CellSegUDA for KIRC->TNBC (Experiment-1) with different set of values of lambda_adv and lambda_recons:
>
>                                                                   Experiment-1
> Lambda_adv     Lambda_recons      IoU%     Dice score
> 0.1                                 0.1                  56.32         0.7168
> 0.1                                 0.01                55.58         0.7102
> 0.1                                 0.001              53.79         0.6976
> 0.01                               0.1                  56.47         0.7173
> 0.01                               0.01                54.79         0.7048
> 0.01                               0.001              54.25         0.7180
> 0.001                             0.1                  58.98         0.7385
> 0.001                             0.01                59.02         0.7394
> 0.001                             0.001              57.25         0.7247
>
> From the above results, we can see that when we set lambda_adv to 0.1 and 0.01, the highest IoU% is 56.32 and 56.47 respectively which decreases with the decrement of lambda_recons. The reason of comparatively lower IoU% is that, larger values of lambda_adv strongly forces the segmentation model to generate target domain predictions similar to source domain predictions. Although our intuition is that cell segmentation outputs are domain-invariant, but it’s still unclear that how much similar or different they are. In our experiments, the best result came out when we chose relatively smaller value (0.001) for lambda_adv. With this value of lamda_adv, setting the value of 0.01 for lambda_recons gave us the best performance. Therefore, we chose this set of values as the weights of lambdas in our experiments.
>
> Comment 2: The paper listed comprehensive prior works, but actually did not compare with them except the basic U-Net and DA-ADV. Only very similar domains are evaluated for this method.
>
> Response to Comment 2: In our paper, we compared our method to DA-ADV (Dong et al) which can be considered as a SOTA and representative of domain adaptation approaches in biomedical image segmentation. Another reason for choosing DA-ADV is that, their approach is close to ours as they also apply domain adaptation at the output space. We also compared our method to U-Net as it is a dominant fully-supervised method for cell and other biomedical image segmentation. We also used U-Net to check how it works when directly applying transfer learning, and getting an experimental lower bound and upper bound. Most of the prior works (discussed in the beginning of the second paragraph of Introduction) are designed and evaluated on completely different domain (for example, real world synthetic and real city images) than biomedical images. However, we apologize for not including more domain adaptation and transfer learning methods in our experiments.

---

### Official Review · AnonReviewer1 · 2020-03-13
**without a clearly defined contribution**

**Rating:** 2
**Confidence:** 4

**Summary:**

To segment cells without labelled data, authors proposed unsupervised deep learning method based on domain adaptation. The proposed method learns to segment instances in the target domain, by learning supervised segmentation in source domain that is regularized with an adversarial loss that keeps the distribution of the segmentation prediction in the target and source domain similar. Additionally, the authors use reconstruction loss to ensure that target predictions spatially correspond to the target images.

**Strengths:**

(1) The method is evaluated on two publicly available cell dataset (KIRC and TNBC) that are interchangeably used as a source or target domain.
(2) The method is evaluated in unsupervised and semi-supervised scenario.

**Weaknesses:**

(1) The references are chaotic, especially those in the second paragraph of the Introduction. There is no explanation of how the papers cited in the manuscript are related to the proposed method, and thus how the authors choose exactly them. The cited papers should lead the reader to the contribution of the manuscript, which is not the case with this manuscript. Furthermore, authors should not use the arXiv version for citation, since all the papers cited in the manuscript were published in the prestige conferences (CVPR, ECCV, … ) or journals (TMI).

(2) Mainly because of the previous point, it is not clear whether the manuscript has a technical contribution. It seems as the proposed architecture is already being used in CV community, but it is not clear whether there is a paper that used all three losses in the same way as authors do. Some similar architectures can also be found tested on cell dataset [1]. Authors should be clear about their contribution.

(3) Did author try to reconstruct not just target domain, but also source domain?

(4) There is no comparison with SOTA. The DA-ADV method by Dong et al. presented on MICCAI 2018 is good work on smegmatic segmentation of lung in X-ray images, but not evaluated on cell segmentation. Moreover, it is not explained how this approach is used for instance segmentation. How Dong et al. method differs from their approach without reconstruction loss, i.e. “CellSegUDA w/o recons”? Did the authors used the original code of Dong et al.? What about the results of other methods (e.g. [1])? Are the results of “U-Net (target-trained)” on these datasets inline with SOTA methods learned in supervised meaner?

(5) Why there is a decrease in the performance from U-Net (source 100% + target 25%) to U-Net (source 100% + target 50%) for more than 4%. I also find unfair writing U-Net (source 100% + target XX%), while CellSegSSDA (XX%).  It would be more consistent if writing also CellSegSSDA (source 100% + target XX%).

(6) What are the results of authors approach on target domain when trained with all source and all target images that are labelled, i.e. CellSegSSDA (source 100% + target 100%)? I would expect the results to be at least as good as U-Net (target-trained).

(7) Comparison of the segmentation results presented in Fig. 4 is difficult. Yellow and blue arrows are sparse and not helpful, mainly due to “previous method” and “following method”.

(8) Following sentence is misleading: “our proposed UDA method, CellSegUDA, outperforms both of a fully-supervised model trained in the source domain, and a baseline UDA model.” It is not clear whether it has been evaluated on the target domain. Thus, CellSegUDA is better then U-Net (source-trained) but not U-Net (target-trained). A better formulation could be: “our proposed UDA method, CellSegUDA, outperforms a fully-supervised model trained on the source domain and evaluated on the target domain.”.


[1] Xing et al., Adversarial Domain Adaptation and Pseudo-Labeling for Cross-Modality Microscopy Image Quantification, MICCAI, 2019

**Justification Of Rating:**

The contribution is not clearly explained in the manuscript and there is no clear distinction from the SOTA methods. Because the connection to SOTA is missing, evaluation of the method performance is hard for interpretation.

**Paper Type:**

both

**Special Issue:**

no

---

> ### Author Response · Authors · 2020-03-27
> **Response to Reviewer1**
>
> We would like to thank you for the positive remarks, valuable comments and feedback. Below we address the points raised:
>
> Comment 1: The references are chaotic, especially those in the second paragraph of the Introduction. There is no explanation of how the papers cited in the manuscript are related to the proposed method, and thus how the authors choose exactly them. The cited papers should lead the reader to the contribution of the manuscript, which is not the case with this manuscript. Furthermore, authors should not use the arXiv version for citation, since all the papers cited in the manuscript were published in the prestige conferences (CVPR, ECCV, … ) or journals (TMI).
>
> Response to Comment 1: We apologize for this confusion regarding chosen references and the contributions of our paper. In the second paragraph of introduction, we first discussed several SOTA domain adaptation papers that mainly focus on real world (for example, city images) semantic segmentation. Then, we discussed several other prominent papers from biomedical domain. Among these papers, Dong et al. can be considered as a SOTA and representative of domain adaptation approaches in biomedical image segmentation. Also, Dong et al. is closest one to us as they also apply domain adaptation at the output space. We listed the contributions of our paper at the end of introduction. However, we will try to remove the ambiguities regarding our contribution in the final version of our paper. We will also fix the arXiv version related issue for citation.

---

> > ### Author Response · Authors · 2020-03-27
> > **Response to Reviewer1**
> >
> > Comment 2: Mainly because of the previous point, it is not clear whether the manuscript has a technical contribution. It seems as the proposed architecture is already being used in CV community, but it is not clear whether there is a paper that used all three losses in the same way as authors do. Some similar architectures can also be found tested on cell dataset [1]. Authors should be clear about their contribution.
> >
> > Response to Comment 2: We apologize again for this confusion. As mentioned before, DA-ADV (Dong et al.) is the closest one to us. The major differences between DA-ADV (Dong) and CellSegUDA w/o recons are as follows:
> >
> > 1. We use dice-coefficient loss as segmentation loss in CellSegUDA w/o recons whereas DA-ADV (Dong) uses cross-entropy loss. We experimented with cross-entropy loss, dice-coefficient loss, and a combination of both losses. By comparing experimental results and visualizing segmentation predictions, we finally found that using only dice-coefficient loss as segmentation loss works considerably better in our proposed model.
> >
> > 2. In our paper, the discriminator is designed to make the distribution of source domain predictions and target domain prediction similar. Our intention is that, cell segmentation outputs are domain-invariant. We did not utilize source domain ground truths while training the discriminator. However, in DA-ADV, the discriminator is designed so that it can distinguish segmentation predictions from ground truths. We also experimented with following this setting for our discriminator but found comparatively poor performance in our case.
> >
> > 3. From network model perspective, both of our segmentation model and discriminator model are different from DA-ADV ones. We chose U-Net as our segmentation model as it is proven to be prominent for cell image segmentation. We also designed the discriminator model ourselves consisting of five convolutional layers. In DA-ADV, FCN and ResNet is used as segmentor and discriminator respectively.
> >
> > 4. Our GAN learning strategy is different from DA-ADV. DA-ADV follows alternative training scheme which does not give any better results in our case. We train our proposed UDA model using jointly training scheme.
> >
> > We extend CellSegUDA w/ reconstruction by proposing CellSegUDA. In CellSegUDA, we added decoder and using reconstructions loss which gives better accuracy. CellSegUDA can predict cells that would have been missed out CellSegUDA w/o reconstruction. Also, CellSegUDA is capable of removing some false positive cells.
> >
> > Finally, we enhance CellSegUDA by proposing CellSegSSDA in which we apply semi-supervision from target domain. This extension would be quite important in medical domain since many of the publicly available biomedical datasets can be found partially annotated.
> >
> > Although the mentioned paper [1] is tested on cell dataset, their work is focused on cell detection. In this paper [1], labelled source domain images are adapted to target domain using cycle-consistent loss, and then the detector learns to locate cells with adapted source images. Finally, the detector is applied to unannotated target training data for generating pseudo-labels, and the detector is fine-tuned using these artificial target domain annotations. The methodology and the architecture of this paper [1] largely differs from ours.
> >
> > References:
> > 1. Xing et al., Adversarial Domain Adaptation and Pseudo-Labeling for Cross-Modality Microscopy Image Quantification

---

> > > ### Author Response · Authors · 2020-03-27
> > > **Response to Reviewer1**
> > >
> > > Comment 3: Did author try to reconstruct not just target domain, but also source domain?
> > >
> > > Response to Comment 3: Yes, we tried to reconstruct also using source domain. Following is the experimental results in which we compare the performance of w/o reconstruction, only source reconstruction, only target reconstruction, and both of source and target reconstruction for experiment-1:
> > >
> > > Method                                                                                                      Experiment-1
> > >                                                                                                                     KIRC -> TNBC
> > >                                                                                                                IoU%     Dice score
> > > CellSegUDA w/o reconstruction                                                      56.56          0.72
> > > CellSegUDA with source reconstruction only                               57.48          0.7274
> > > CellSegUDA with target reconstruction only                                59.02          0.7394
> > > CellSegUDA with both of source and target reconstruction      58.70          0.7373
> > >
> > > From the above results, we see that CellSegUDA with only target reconstruction performs best . This result is intuitive as including source domain in reconstruction loss will help the decoder to learn to transform source domain predictions to source images. However, there is a domain shift between source images and target images, and we are particularly interested in learning the decoding in the target domain to make target domain predictions correlated to their corresponding images in target domain. Because of this, we only included target domain while calculating the reconstruction loss in our paper.
> > >
> > > Comment 4: There is no comparison with SOTA. The DA-ADV method by Dong et al. presented on MICCAI 2018 is good work on smegmatic segmentation of lung in X-ray images, but not evaluated on cell segmentation. Moreover, it is not explained how this approach is used for instance segmentation. How Dong et al. method differs from their approach without reconstruction loss, i.e. “CellSegUDA w/o recons”? Did the authors used the original code of Dong et al.? What about the results of other methods (e.g. [1])? Are the results of “U-Net (target-trained)” on these datasets inline with SOTA methods learned in supervised meaner?
> > >
> > > Response to Comment 4: Dong et al. (DA-ADV) can be considered as a SOTA and representative of domain adaptation approaches in biomedical image segmentation. Also, DA-ADV is closest one to us as they also apply domain adaptation at the output space. Other than DA-ADV, we also compared our method to U-Net as it is a dominant fully-supervised method for cell and other biomedical image segmentation. We also used U-Net to check how it works when directly applying transfer learning, and getting an experimental lower bound and upper bound.
> > >
> > > Comment 5: Why there is a decrease in the performance from U-Net (source 100% + target 25%) to U-Net (source 100% + target 50%) for more than 4%. I also find unfair writing U-Net (source 100% + target XX%), while CellSegSSDA (XX%).  It would be more consistent if writing also CellSegSSDA (source 100% + target XX%).
> > >
> > > Response to Comment 5: We will modify “CellSegSSDA (XX%)” to “CellSegSSDA (source 100% + target XX%)” in the final version. We would like to thank you for pointing this out. It is still unclear to us why performance decreases from U-Net (source 100% + target 25%) to U-Net (source 100% + target 50%).

---

> > > > ### Author Response · Authors · 2020-03-27
> > > > **Response to Reviewer1**
> > > >
> > > > Comment 6: What are the results of authors approach on target domain when trained with all source and all target images that are labelled, i.e. CellSegSSDA (source 100% + target 100%)? I would expect the results to be at least as good as U-Net (target-trained).
> > > >
> > > > Response to Comment 6: Following is the result when we train our model with all labelled source and target images, i.e. CellSegSSDA (source 100% + target 100%):
> > > >
> > > > Method                                                                            Experiment-1                       Experiment-2
> > > >                                                                                       IoU%    Dice Score                IoU%    Dice Score
> > > > CellSegSSDA (source 100% + target 100%)	      65.96       0.7942                   61.46        0.7577
> > > > U-Net (target-trained)                                               66.57       0.7985                   62.04        0.7621
> > > >
> > > > These results are very close to the result of U-Net (target-trained). Target-trained U-Net only learns to generate predictions specifically for the target domain only. On contrary, CellSegSSDA (source 100% + target 100%) learns to generate predictions for both domains due to the segmentation loss. At the same time, it learns to generate target domain predictions which look similar to the source domain prediction using adversarial loss. Our intuition is that, due to generating source-domain lookalike predictions even for fully annotated target domain, CellSegSSDA (source 100% + target 100%) could not outperform the U-Net (target-trained).
> > > >
> > > > Comment 7: Comparison of the segmentation results presented in Fig. 4 is difficult. Yellow and blue arrows are sparse and not helpful, mainly due to “previous method” and “following method”.
> > > >
> > > > Response to Comment 7: We apologize for this difficulty in understating Fig. 4. In fact, visually differentiating among various cell segmentation outputs is a tough task for which we added arrows. However, we will try our best to make this visualization clearer in the final version.
> > > >
> > > > Comment 8: Following sentence is misleading: “our proposed UDA method, CellSegUDA, outperforms both of a fully-supervised model trained in the source domain, and a baseline UDA model.” It is not clear whether it has been evaluated on the target domain. Thus, CellSegUDA is better then U-Net (source-trained) but not U-Net (target-trained). A better formulation could be: “our proposed UDA method, CellSegUDA, outperforms a fully-supervised model trained on the source domain and evaluated on the target domain.”.
> > > >
> > > > Response to Comment 8: We apologize for this ambiguity. This will be corrected in the final version.

---

> > > > > ### Comment · AnonReviewer1 · 2020-04-03
> > > > > **further investigation of the baseline method is needed**
> > > > >
> > > > > If only $L_{recon}$ is to blame from the discrepancy in the results then the results of U-Net (source 100% + target 100%), should be at least as good as U-Net (target-trained). Is that the case?
> > > > >
> > > > > Authors should include the results of CellSegSSDA (source 100% + target 100%) and U-Net (source 100% + target 100%) into the table and comment the results.

---

> > > > > > ### Author Response · Authors · 2020-04-04
> > > > > > **Response to Reviewer1**
> > > > > >
> > > > > > Following table shows the results of U-Net (source 100% + target 100%), CellSegSSDA (source 100% + target 100%), and U-Net (target-trained) for both experiments:
> > > > > >
> > > > > > Method                                                                            Experiment-1                              Experiment-2
> > > > > >                                                                                      IoU%     Dice Score                     IoU%     Dice Score
> > > > > > U-Net (source 100% + target 100%)                      61.70        0.7601                         61.35        0.7561
> > > > > > CellSegSSDA (source 100% + target 100%)          65.96        0.7942                         61.46        0.7577
> > > > > > U-Net (target-trained)                                              66.57        0.7985                         62.04        0.7621
> > > > > >
> > > > > > We see that, U-Net (source 100% + target 100%) gives comparatively poor performance for both experiments. The reason is that, there is a domain shift between source domain training images and target domain training images. Unfortunately, learning a more generalized (domain-invariant) model by optimizing only segmentation loss does not work well in cell segmentation problem. In the case of CellSegSSDA (source 100% + target 100%), we consider adversarial loss (target domain) and reconstruction loss (target domain) in addition to segmentation loss. And, for U-Net (target-trained), we train it to optimize the segmentation loss only for the target domain.

---

> > > > ### Comment · AnonReviewer1 · 2020-04-03
> > > > **SOTA results are still unclear.**
> > > >
> > > > The authors did not provide an answer to my question: Are the results of “U-Net (target-trained)” on these datasets inline with SOTA methods learned in supervised meaner?

---

> > > > > ### Author Response · Authors · 2020-04-04
> > > > > **Response to Reviewer1**
> > > > >
> > > > > We apologize for not providing the answer to this question in the rebuttal. Following is our answer to this question:
> > > > >
> > > > > Yes, the results of U-Net (target-trained) on these datasets are inline with SOTA methods learned in supervised manner. For TNBC dataset, we compared our U-Net (target-trained) result with SOTA methods like FCN, SegNet, Mask-RCNN, DCAN, Mirco-Net, DIST, HoVer-Net and NB model [1, 2, 3]. Similarly, for KIRC dataset, our U-Net (target-trained) result is compared with SOTA methods FCN and SegNet [4].
> > > > >
> > > > > References:
> > > > > 1. Graham et al., HoVer-Net: Simultaneous Segmentation and Classification of Nuclei in Multi-Tissue Histology Images, Medical Image Analysis 2019
> > > > > 2. Yoo et al., PseudoEdgeNet: Nuclei Segmentation only with Point Annotations, MICCAI 2019
> > > > > 3. Cui et al., A Deep Learning Algorithm for One-step Contour Aware Nuclei Segmentation of Histopathological Images, Med Biol Eng Comput 2019
> > > > > 4. https://rc.library.uta.edu/uta-ir/handle/10106/26395

---

### Official Review · AnonReviewer3 · 2020-03-13
**Nice paper**

**Rating:** 3
**Confidence:** 5
**Recommendation:** Poster

**Summary:**

Paper presents a domain adaptation method for the segmentation of cell segmentation.
The motivation of the method is based on the fact that the ground truth labels are domain invariant so the model is trained to produce outputs for the target data that look like those from the source data.

The authors run experiments in unsupervised domain adaptation settings and semi-supervised domain adaptation. I particularly liked that the authors compared how their method performs compared to the baselines when different degrees of labeled data is accessible.

**Strengths:**

The motivation is good,
The paper is very clear: The loss functions and different component of the model is well defined.
The experiments or bi-directional (Domain A to B and B to A).
Comparision with a baseline is done
Some ablation study is done.


**Weaknesses:**

-I wish the authors would include more domain adaptation and transfer learning baselines.
-Also, it would have been better if more datasets were considered.
-There are some typos in the paper but are not major issues.
-There is no comparison with other methods on this dataset.

**Justification Of Rating:**

As mentioned before, I see this paper to be technically sound and the experiments support the motivation and the hypothesis of the paper.
I think this paper would be a good addition to the conference.

**Paper Type:**

validation/application paper

**Questions To Address In The Rebuttal:**

did you try adding reconstruction loss for the source images as well?


**Special Issue:**

no

---

> ### Author Response · Authors · 2020-03-27
> **Response to Reviewer3**
>
> We would like to thank you for the positive remarks, valuable comments and feedback. Below we address the points raised.
>
> Question 1: did you try adding reconstruction loss for the source images as well?
>
> Response to Question 1: Yes, we tried adding reconstruction loss for source images. Following is the experimental results in which we compare the performance of w/o reconstruction, only source reconstruction, only target reconstruction, and both of source and target reconstruction for experiment-1:
>
> Method                                                                                                         Experiment-1
>                                                                                                                         KIRC->TNBC
>                                                                                                                   IoU%        Dice score
> CellSegUDA w/o reconstruction                                                          56.56             0.72
> CellSegUDA with source reconstruction only                                   57.48             0.7274
> CellSegUDA with target reconstruction only                                    59.02             0.7394
> CellSegUDA with both of source and target reconstruction          58.70             0.7373
>
> From the above results, we see that CellSegUDA with only target reconstruction performs best. This result is intuitive as including source domain in reconstruction loss will help the decoder to learn to transform source domain predictions to source images. However, there is a domain shift between source images and target images, and we are particularly interested in learning the decoding in the target domain to make target domain predictions correlated to their corresponding images in target domain. Because of this, we only included target domain while calculating the reconstruction loss in our paper.
>
> Comment 1: I wish the authors would include more domain adaptation and transfer learning baselines.
>
> Response to Comment 1: In our paper, we compared our method to DA-ADV (Dong et al) [1] which can be considered as a SOTA and representative of domain adaptation approaches in biomedical image segmentation. Another reason for choosing DA-ADV is that, their approach is close to ours as they also apply domain adaptation at the output space. We also compared our method to U-Net as it is a dominant fully-supervised method for cell and other biomedical image segmentation. We also used U-Net to check how it works when directly applying transfer learning, and getting an experimental lower bound and upper bound. Most of the prior works (discussed in the beginning of the second paragraph of Introduction) are designed and evaluated on completely different domain (for example, real world synthetic and real city images) than biomedical images. However, we apologize for not including more domain adaptation and transfer learning methods in our experiments.
>
> Comment 2: Also, it would have been better if more datasets were considered.
>
> Response to Comment 2: Unfortunately, there are not much publicly available fully annotated cell segmentation datasets. However, other than KIRC and TNBC datasets, we collected two more datasets for which source organ is unknown to us. We ran experiments with those two datasets and achieved the superiority of our method compared to DA-ADV and U-Net. We could not include those experiments due to the page limit of the submitted paper.
>
> Comment 3: There are some typos in the paper but are not major issues.
>
> Response to Comment 3: We apologize for the typos. We will correct the typos in the final version.
>
> References:
> 1. Dong et al., Unsupervised Domain Adaptation for Automatic Estimation of Cardiothoracic Ratio

---

### Official Review · AnonReviewer4 · 2020-03-14
**Modifications on the classical UDA with adversarial by adding a reconstruction loss and replacing Cross Entropy by DiceLoss**

**Rating:** 3
**Confidence:** 4

**Summary:**

This paper proposes modifications on the classical UDA with adversarial by adding a reconstruction loss, which is motivated by the application, histology images, where segmentation output masks and input images show similarities.
Then, the classical Cross Entropy in the source domain is replaced by a Dice Loss . The experiments are done on histology images.

**Strengths:**

- The paper is clear and easy to follow. The ideas are straightforwards.
- An unsupervised as well as semi -supervised frameworks are explored, with a growing number of training examples.
- Experiments are done on the 2 adaptation directions, on histology images.
- Figures are nice.


**Weaknesses:**


- The novelty of the paper compared to many classical UDA works[1,2], i.e. the introduction of the reconstruction loss, isn't made clear enough.
- Although clear, the paper could be more concise.
- The models using semi-supervision could have been compared to Dong + semi-supervision as well, to compare with another DA method.
- Improvement yielded by the reconstruction loss is actually limited
- Some tipos (introduction)

1] Dong et al., Unsupervised Domain Adaptation for Automatic Estimation of Cardiothoracic Ratio
2] Tsai et al., Learning to Adapt Structured Output Space for Semantic Segmentation

**Justification Of Rating:**

The paper is well written, and is easy to follow. Novelty is limited, and could be better discussed/ clarified.
The improvent yield by the reconstruction loss is limited. Further, in the SSDA framework, no comparison has been made with a method actually using a domain adaptation technique.

**Paper Type:**

both

**Questions To Address In The Rebuttal:**

- What is the difference between DA-ADV (Dong) and CellSegUDA w/o recons ? Is it just the replacement of the Cross Entropy Loss by the Dice Loss ? This should be clarified.
- The improvement in the metrics is quite small, especially between CellSegUDA w/o and w/ reconstruction, which is the main novelty of the paper. Are these improvements significative ? Seeing the standard deviation for the metrics could be helpful.


**Special Issue:**

no

---

> ### Author Response · Authors · 2020-03-27
> **Response to Reviewer4**
>
> We would like to thank you for the positive remarks, valuable comments and feedback. Below we address the points raised:
>
> Question 1: What is the difference between DA-ADV (Dong) and CellSegUDA w/o recons? Is it just the replacement of the Cross Entropy Loss by the Dice Loss? This should be clarified.
>
> Response to Question 1: We apologize for not clarifying the contributions in our paper. The major differences between DA-ADV (Dong et al) [1] and CellSegUDA w/o recons are as follows:
>
> 1. We use dice-coefficient loss as segmentation loss in CellSegUDA w/o recons whereas DA-ADV (Dong) uses cross-entropy loss. We experimented with cross-entropy loss, dice-coefficient loss, and a combination of both losses. By comparing experimental results and visualizing segmentation predictions, we finally found that using only dice-coefficient loss as segmentation loss works considerably better in our proposed model.
>
> 2. In our paper, the discriminator is designed to make the distribution of source domain predictions and target domain prediction similar. Our intention is that, cell segmentation outputs are domain-invariant. We did not utilize source domain ground truths while training the discriminator. However, in DA-ADV, the discriminator is designed so that it can distinguish segmentation predictions from ground truths. We also experimented with following this setting for our discriminator but found comparatively poor performance in our case.
>
> 3. From network model perspective, both of our segmentation model and discriminator model are different from DA-ADV ones. We chose U-Net as our segmentation model as it is proven to be prominent for cell image segmentation. We also designed the discriminator model ourselves consisting of five convolutional layers. In DA-ADV, FCN and ResNet is used as segmentor and discriminator respectively.
>
> 4. Our GAN learning strategy is different from DA-ADV. DA-ADV follows alternative training scheme which does not give any better results in our case. We train our proposed UDA model using jointly training scheme.
>
> We extend CellSegUDA w/ reconstruction by proposing CellSegUDA. In CellSegUDA, we added decoder and using reconstructions loss which gives better accuracy. CellSegUDA can predict cells that would have been missed out CellSegUDA w/o reconstruction. Also, CellSegUDA is capable of removing some false positive cells. Finally, we enhance CellSegUDA by proposing CellSegSSDA in which we apply semi-supervision from target domain. This extension would be quite important in medical domain since many of the publicly available biomedical datasets can be found partially annotated.
>
> Question 2: The improvement in the metrics is quite small, especially between CellSegUDA w/o and w/ reconstruction, which is the main novelty of the paper. Are these improvements significative ? Seeing the standard deviation for the metrics could be helpful.
>
> Response to Question 2: CellSegUDA w/ reconstruction gives 2.46 higher IoU% than CellSegUDA w/o reconstruction in the first experiment. Although this improvement seems limited numerically, the addition of decoder and reconstruction loss helps to segment some missing cells and remove false positive cells which is illustrated in Fig. 3 and Fig 4 from experiment-1 and experiment-2 respectively. We apologize that we could not include more visualizations in our paper which would similarly show the effectiveness of CellSegUDA over CellSegUda w/o reconstruction.
>
> Following table shows the standard deviation for the metrics for both experiments:
>
> Method                                                 Experiement-1                                          Experiment-2
> 	                                          STD of IoU%    STD of Dice score	     STD of IoU%        STD of Dice score
> CellSegUDA w/o recons	       0.0425                  0.0339                        0.0648                      0.0538
> CellSegUDA w/ recons	       0.0469                  0.0365                        0.0558                      0.0456
>
> References:
> 1. Dong et al., Unsupervised Domain Adaptation for Automatic Estimation of Cardiothoracic Ratio

---

> > ### Author Response · Authors · 2020-03-27
> > **Response to Reviewer4**
> >
> > Comment 1: The models using semi-supervision could have been compared to Dong + semi-supervision as well, to compare with another DA method.
> >
> > Response to Comment 1: We would like to thank you for this suggestion. Following is the experimental result of Dong + semi-supervision and comparison to CellSegSSDA for KIRC->TNBC (Experiment-1) which shows the excellence of CellSegSSDA. We are thinking to include this result in the final version.
> >
> > Method                                                              Experiment-1
> >                                                                              KIRC->TNBC
> >                                                                       IoU%            Dice score
> > Dong + semi-supervision (10%)              58.63               0.7370
> > CellSegSSDA (10%)                                    60.96               0.7557
> > Dong + semi-supervision (25%)              59.15               0.7439
> > CellSegSSDA (25%)                                    62.94               0.771
> > Dong + semi-supervision (50%)              59.76               0.7504
> > CellSegSSDA (50%)                                    63.59               0.7748
> > Dong + semi-supervision (75%)              60.34               0.7586
> > CellSegSSDA (75%)                                    64.96               0.7862
> >
> > Comment 2: Although clear, the paper could be more concise. Some typos (introduction).
> >
> > Response to Comment 2: We apologize for the typos. We will correct the typos and try to make the paper more concise in its final version.

---

### Comment · Area_Chair1 · 2020-03-29
**Discussion**


Dear reviewers,

We are now starting the discussion phase for MIDL.  So far, this paper has received three weak accepts and one weak reject. Please check the author responses and participate to the discussions for confirming your final evaluations. Thank you.

---

### Meta-Review · Area_Chair1 · 2020-04-05
**MetaReview of Paper336 by AreaChair1**

**Rating:** 3
**Recommendation For Accepted Papers:** Poster

**Metareview:**

Three out of four reviewers recommended "weak accept", being convinced by the value of application on histology images, bi-directional evaluation of the adaptation method and clear presentation of this paper. I also think domain adaptation is an important problem to be tackled in the field of digital histopathology. This paper will add good value to this research topic.  The final version should address the questions mentioned by the reviewers.

**Paper Type:**

both

**Special Issue:**

no

---

> ### Author Response · Authors · 2020-04-08
> **Response to Area Chair**
>
> We would like to thank the Area Chair for positive feedback and valuable comments. We will address the questions mentioned by the reviewers in the final version.

---

### Decision · Program_Chairs · 2020-04-11

Accept